# Untargeted Metabolic Profiling of Extracellular Vesicles of SARS-CoV-2-Infected Patients Shows Presence of Potent Anti-Inflammatory Metabolites

**DOI:** 10.3390/ijms221910467

**Published:** 2021-09-28

**Authors:** Faisal A. Alzahrani, Mohammed Razeeth Shait Mohammed, Saleh Alkarim, Esam I. Azhar, Mohammed A. El-Magd, Yousef Hawsawi, Wesam H. Abdulaal, Abdulaziz Yusuf, Abdulaziz Alhatmi, Raed Albiheyri, Burhan Fakhurji, Bassem Kurdi, Tariq A. Madani, Hassan Alguridi, Roaa S. Alosaimi, Mohammad Imran Khan

**Affiliations:** 1King Fahd Medical Research Center, Embryonic Stem Cells Unit, Department of Biochemistry, Faculty of Science, King Abdulaziz University, Jeddah 21589, Saudi Arabia; skarim@kau.edu.sa (S.A.); a.zi@windowslive.com (A.Y.); mr.abdulaziz1415@hotmail.com (A.A.); dr.alguridi@gmail.com (H.A.); 2Centre of Artificial Intelligence in Precision Medicines (CAIPM), King Abdulaziz University, Jeddah 21589, Saudi Arabia; razeeth.new@gmail.com (M.R.S.M.); whabdulaal@kau.edu.sa (W.H.A.); 3King Fahd Medical Research Center, Special Infectious Agents Unit, Medical Laboratory Technology Department, Faculty of Applied Medical Sciences, King Abdulaziz University, Jeddah 21589, Saudi Arabia; eazhar@kau.edu.sa; 4Department of Anatomy & Embryology, Faculty of Veterinary Medicine, Kafrelsheikh University, Kafrelsheikh 33516, Egypt; mohamed.abouelmagd@vet.kfs.edu.eg; 5Research Center, King Faisal Specialist Hospital and Research Center, P.O. Box 40047, Jeddah 21499, Saudi Arabia; hyousef@kfshrc.edu.sa; 6King Fahd Medical Research Center, Cancer and Mutagenesis Unit, Department of Biochemistry, Faculty of Science, King Abdulaziz University, Jeddah 21589, Saudi Arabia; 7Department of Biological Sciences, Faculty of Science, King Abdulaziz University, Jeddah 21589, Saudi Arabia; ralbiheyri@kau.edu.sa (R.A.); dr.burhan@igene-sa.com (B.F.); 8iGENE Center, Jeddah 23484, Saudi Arabia; 9Department of Pediatrics, Faculty of Medicine, King Abdulaziz University, Jeddah 21589, Saudi Arabia; bskurdi@kau.edu.sa; 10Department of Medicine, Faculty of Medicine, King Abdulaziz University, P.O. Box 80215, Jeddah 21589, Saudi Arabia; tmadani@kau.edu.sa; 11Jeddah Regional Laboratory, Molecular Biology Department, Ministry of Health, Jeddah 22421, Saudi Arabia; 12East Jeddah General Hospital, Ministry of Health, Jeddah 22253, Saudi Arabia; roalosaimi@moh.gov.sa

**Keywords:** extracellular vesicles, metabolomics, COVID-19, 7-α,25-Dihydroxycholesterol, 15-d-PGJ2

## Abstract

Extracellular vesicles (EVs) carry important biomolecules, including metabolites, and contribute to the spread and pathogenesis of some viruses. However, to date, limited data are available on EV metabolite content that might play a crucial role during infection with the SARS-CoV-2 virus. Therefore, this study aimed to perform untargeted metabolomics to identify key metabolites and associated pathways that are present in EVs, isolated from the serum of COVID-19 patients. The results showed the presence of antivirals and antibiotics such as Foscarnet, Indinavir, and lymecycline in EVs from patients treated with these drugs. Moreover, increased levels of anti-inflammatory metabolites such as LysoPS, 7-α,25-Dihydroxycholesterol, and 15-d-PGJ2 were detected in EVs from COVID-19 patients when compared with controls. Further, we found decreased levels of metabolites associated with coagulation, such as thromboxane and elaidic acid, in EVs from COVID-19 patients. These findings suggest that EVs not only carry active drug molecules but also anti-inflammatory metabolites, clearly suggesting that exosomes might play a crucial role in negotiating with heightened inflammation during COVID-19 infection. These preliminary results could also pave the way for the identification of novel metabolites that might act as critical regulators of inflammatory pathways during viral infections.

## 1. Introduction

Severe acute respiratory syndrome coronavirus 2 (SARS-CoV-2) is a recently detected coronavirus, which resulted in the pandemic spread of Corona Virus Disease 2019 (COVID-19). Although its genome sequence is somewhat close to SARS-CoV, the novel virus has improved infectivity and hidden initial clinical signs, hindering early detection of suspected cases and control of SARS-CoV-2 transmission [1,2]. Some researchers attributed these enhanced features to the high-affinity binding of SARS-CoV-2 spike protein (S) to its receptor, angiotensin-converting enzyme 2 (ACE2), attributed to the presence of several mutations in the receptor-binding domain (RBD) of S, especially the novel furin cleavage insert, which is unique to this virus [3,4,5,6,7,8,9,10]. However, these mutations alone may not explain the higher contagious spread of SARS-CoV-2, and there may be other mechanisms that enhance infectivity and escape from host immune responses.

Extracellular vesicles (EVs) are a heterogeneous population of membranous structures released by cells into outer space under both normal and pathophysiologic conditions [11]. The majority of EVs range in diameter from 30 to 100 nanometers and include exosomes from the endosomal compartment and ectosomes from the plasma membrane. EVs are known to be secreted by most of the systemic cell types [12] and can be abundantly found in human serum [13]. EVs contain a wide range of biological molecules, such as DNA, RNA, miRNA, proteins, functional enzymes, cholesterol and sphingomyelin, and metabolites [14]. The lipid bilayer of EVs enables stability of internal cargo molecules while EVs circulate through body fluids. As a result, EVs can transfer their specific cargo to various target cells as a means of regulating target cell physiological and pathological processes, including host immune responses. The specificity of cargo molecules highly depends on the cell type of origin and circumstances of EV release, such as stress, infections, and drug exposures [15,16]. However, mechanisms for packaging of specific cargo into EVs remain elusive.

EVs are under intense investigation, as they play a significant role in disease and simultaneously possess the potential for diagnostics as biomarkers of pathological conditions such as cardiovascular disease, Alzheimer’s disease, cancer, and viral infection [13,14,17,18]. Biogenesis of exosomes shares some cellular components (such as endosomes, multivesicular bodies, and lysosomes) that are hijacked by viruses after entrance into the cell [17]. Concerning their role in viral disease pathogenesis, EVs such as exosomes can transmit nucleic acids and proteins between viral infected and uninfected cells, thereby facilitating cell-to-cell viral transmission and regulating host immune responses. Several viruses such as HBV, HCV, and RSV utilize EVs to infect host cells [13,14,17,18,19]. Exosomes derived from a cell culture system and patients’ sera with complete replication of the HCV genome showed an ability to transmit infection to normal human hepatocytes [18,20]. Moreover, exosomes can activate immune response through induction of cytokine release during cytokine storm associated with respiratory syncytial virus (RSV) infection [17]. Interestingly, the exosomes isolated from the blood of hepatitis B patients contained protein and nucleic acids of HBV, and exosomal contents were also detected in the natural killer cells (NK) and contributed to the suppression of immunity [21]. On the other hand, SARS-CoV was reported to use EVs during infection of AT2 cells [22]. SARS-CoV-2 products were detected in EVs of renal cells of COVID-19 patients [23,24], although other studies did not find SARS-CoV-2 RNA in EVs [25,26,27]. Exosomes can transmit ACE2 between host cells [28], making recipient cells potentially more prone to virus infection [29]. Viral particles within EVs were reliably detected in samples originated from the patients infected with SARS-CoV-2 [30,31,32,33]. 

Metabolomics provide a unique opportunity for identifying novel biomarkers and biochemical pathways to enhance early detection of disease, providing pathological insights beyond traditional “omics” methods. Quantitative measurement of circulating small-molecule metabolites that may participate in disease pathogenesis [34] may improve diagnosis, unravel appropriate therapeutic targets, and enable a more precise prognosis of disease outcome [35]. Metabolomics studies have successfully identified biomarkers for diagnosis, progression, and treatment response for age-related diseases such as cancer, diabetes, and autoimmune diseases [36,37,38,39]. A recent study has reported the presence of an abundant amount of coagulation-related protein Kininogen-1 in EVs, such as exosomes isolated from COVID-19 patients, and suggested that exosomes may act as a reserve of Kininogen-1 [40]. As for EVs and metabolites, GM3-enriched exosomes are associated with COVID-19 severity and progression [41]. Exosomal C-reactive protein (CRP) was detected in the plasma of COVID-19 patients and was positively related to disease severity, suggesting a crucial role for exosomes in CRP transport among host cells [40]. Exosomes from plasma of severe cases (ICU) of COVID-19 had an increased level of the fibrosis markers tenascin-C (TNC) and fibrinogen-β (FGB) [25]. Additionally, exosome-based tactics were also recommended to handle COVID-19 [29] or inhibit SARS-CoV-2 infection [42] or as carriers to provide protease inhibitors to combat COVID-19 [43].

Knowing the role of the EVs such as exosomes in SARS-CoV-2 infection could solve some mysterious gaps related to COVID-19 pathogenesis and open new opportunities for the development of effective treatments. The aforementioned data indicate that EVs could play a crucial role in the transmission and pathogenesis of some viruses, such as HIV, HBV, HCV, and RSV. However, to date, limited data are available on EV content/structure and role during infection with SARS-CoV-2. Moreover, the roles of EVs in immune suppression and regulation, antigen presentation, and inflammation have not yet been fully elucidated. In the current work, we have used isolated EVs to perform untargeted metabolomics to identify key metabolites and associated pathways that are present in COVID-19 patients and could be developed as a therapeutic/diagnostic target.

## 2. Results

### 2.1. Demographics of the Study Population

The study population included nine subjects equally allocated into the following three groups: control (uninfected), mild to severe COVID-19 cases, and severe-ICU COVID-19 cases (Table 1). The mean ages of control subjects, mild–severe, and severe-ICU COVID-19 patients were 30.33 ± 2.40, 54.33 ± 3.93, and 59.33 ± 7.69 years, respectively. Table 1 also shows the main medication used to treat each COVID-19 patient.

### 2.2. Characterization of Isolated Exosomes

TEM examination for EVs extracted from the serum of COVID-19 patients and healthy controls revealed the presence of nanovesicles of different sizes (Figure 1A,B). The size of these exosomes ranged from 90 to 250 nm in diameter, as detected by NanoSight (Figure 1C).

### 2.3. Metabolomics of COVID-19 Patients’ and Healthy Individuals’ EVs

To assess whether EV-associated metabolites correlate with various traits of COVID-19, we performed untargeted metabolomics in isolated EVs of COVID-19 patients and compared it with healthy individuals by using LC-MS/MS. After the initial runs, all the raw files were analyzed by using XCMS software for feature detection and their total ion chromatograms were merged and visualized (Figure 2A).

After analyzing the feature peaks, a total of 163 rationally adjusted circulating metabolite features were identified in the EV concentrates by using the ESI^+^ mode (Appendix A). Further, two-dimensional principal component analysis (2D-PCA) model score plots of all samples showed no outliers and simultaneously revealed a significant difference in metabolites between the COVID 19 patients and healthy control individuals (Figure 2B).

### 2.4. Differential Metabolites in COVID-19 EVs

Among the total number of rationally adjusted EV metabolites in this study, 95 metabolites were found to be significantly (*p* < 0.05) modulated: either up- or down-regulated in COVID-19 patients when compared with healthy controls. Among these identified metabolites, 29 (upregulated) and 66 (downregulated) were found to be statistically significant in COVID-19 patients’ EVs when compared with healthy control EVs (Appendix A). Next, we plotted metabolites based on self-correlations by using the total number of rationally adjusted metabolites and found a significant correlation coefficient among the identified metabolites (Figure 3A). Further, using the total number of rationally adjusted metabolites, we created a hierarchical cluster analysis (HCA) heatmap and found that the COVID-19 patient samples were well-clustered separately from healthy controls (Figure 3B).

### 2.5. Pathway Enrichment Analysis of Identified Metabolites in EVs

To identify pathways that are associated with identified metabolites in EVs of COVID-19 patients, we used the well-established MetaboAnalystR version 3.0. Results showed that EV-associated metabolites were associated with many important metabolic pathways, such as the glucose-alanine cycle, methyl histidine metabolism, spermidine, and spermine biosynthesis, gluconeogenesis, phosphatidylcholine biosynthesis, alanine, and sphingolipid metabolism (Figure 4A). We further extended to build a lipid-based pathway that was identified in EVs. As shown in Figure 4B, different classes of lipids were enriched, and among them the highest were the fatty acids and conjugates, suggesting their pivotal role in COVID-19 pathophysiology.

Upon segregating the statistically significant metabolites that are upregulated, we noticed that EVs from COVID-19 carry a variety of antivirals and antibiotics, such as Foscarnet, Indinavir, and lymecycline, that were almost absent in the control EVs (Figure 5A). We found increased levels of Stearoylglycine, an acylglycine with C-18 fatty acid group, as the acyl moiety. The acylglycines class of fatty acid metabolites associated with patients with various fatty acid oxidation disorders. Further, we observed significantly high levels of oleic acid (fatty acid) in EVs of COVID-19 patients when compared to healthy EVs (Figure 5B). Next, we observed that COVID-19 EVs contain increased levels of conjugated bilirubin metabolites, namely bilirubin glucuronide and estradiol acetate glucuronide, when compared to EVs of healthy individuals. Surprisingly, we detected an increased amount of Coproporphyrin III (a porphyrin metabolite arising from heme synthesis), known to be elevated during viral infections, in EVs of COVID-19 patients (Figure 5C).

In addition to the above-mentioned results, we noticed induction of three major metabolites that are directly related to the resolution of various inflammatory pathways. We observed increased levels of LysoPS (18:1, a lysophospholipid), 7-α,25-Dihydroxycholesterol (an endogenous ligand for EBI2 receptor on immune cells), and 15-d-PGJ2 (a ligand for the nuclear receptor PPARγ) (Figure 6).

Next, we segregated the statistically significant downregulated EV-associated metabolites and identified thromboxane (a coagulant factor), elaidic acid (an unsaturated fatty acid that has been shown to modulate inflammatory responses), and palmitoylcholine (a phospholipid that plays a crucial role in cell signaling and communication). Further, we observed a significant decline in 20-Dihydrodydrogesterone (DHD), known to modulate endothelial nitric oxide, in EVs of COVID-19 patients when compared to EVs of healthy controls (Figure 7). In addition, we also noticed a significant reduction in many metabolites whose systemic and physiological impacts are still not well-known (Appendix A).

### 2.6. Screening for Identification of Potential COVID-19-Specific Metabolites in EVs

Finally, we sought to learn whether any metabolite or group of metabolites might be considered as COVID-19-specific. For this, we performed the variable importance in projection (VIP) score, and 15 EV metabolites were found to be responsive towards the variation between COVID-19 patient samples and healthy control samples (Figure 8). Among the metabolites that discriminated COVID-19 patient samples from healthy controls were four upregulated metabolites, namely oleic acid, 1,26-hexacosanediol, longamide, and dictagymnin, belonging to different metabolic pathways and suggesting a clear alteration in the respective pathways.

Similarly, downregulated metabolites DL-cerebronic acid, cycloartanol, behenic acid, oleic acid, and dotriacontane were only associated with EVs of COVID-19 patient samples. Based on these results, we have identified some potential candidate metabolites in EVs that can clearly discriminate COVID-19 patients from healthy controls and may be COVID-19-specific metabolites.

## 3. Discussion

EVs such as exosomes can transport molecules derived from viruses, and some reports suggest that they regulate host immune response in a way that enables the virus to evade the immune system [44,45]. Recently, some viral RNAs were identified in exosomes derived from COVID-19 patients, suggesting the possibility of using these exosomes in SARS-CoV-2 transmission within the host [40]. Moreover, exosomal microRNAs were found to play an important role in the formation of thrombi in COVID-19 patients [6]. Several recent studies addressed the ability of exosomes to transmit proteins and metabolites that could participate in virus transmission and severity [25,40,41,42,43]. However, little is known regarding the potential of EVs as carriers for anti-inflammatory and anti-viral metabolites and drugs with a prospective therapeutic role for EVs. Therefore, this study was conducted to unveil such therapeutic potential. Based on the results of the current work, we found: (1) that EVs from COVID-19 patients showed the presence of a variety of antivirals and antibiotics, such as Foscarnet, Indinavir, and lymecycline, (2) increased levels of anti-inflammatory metabolites such as LysoPS, 7-α,25-Dihydroxycholesterol, and 15-d-PGJ2 in EVs from the sera of COVID-19 patients, and (3) decreased levels of metabolites associated with coagulation, such as thromboxane and elaidic acid, in exosomes from the sera of COVID-19 patients.

Our results postulated that sera exosomes orchestrated a regulatory mechanism for modulating the inflammatory storm and thrombosis during SARS-CoV-2 infection. We managed to identify a large pool of metabolites with a wide range of physiological actions in both healthy and disease conditions. However, here, we chose to focus only on the inflammatory pathway-associated metabolites, since inflammation plays a major role in COVID-19-related pathogenesis. Majority of the top enriched metabolites (95 metabolites (*p* < 0.05) modulated, either up- or down-regulated in COVID-19 patients when compared with healthy controls) were still not associated with the pathogenesis of COVID-19, clearly suggesting that future studies need to explore their exact role in the pathogenesis of COVID-19.

Firstly, we noticed that EVs from COVID-19 carry a variety of antivirals and antibiotics, such as Foscarnet, Indinavir, and lymecycline, that were almost absent in the control EVs. It is highly speculative that EVs are used to deliver these drugs to the target cells or organs, primarily to restrain COVID-19 pathogenesis. To the best of our knowledge, we are the first to report such findings related to exosomes in COVID-19 patients. This finding seems to be highly relevant as EVs are known to serve as natural carriers for small molecules, drugs, and vaccine candidates for therapy [46,47,48,49]. However, further research work will provide a clearer picture of the above-mentioned findings.

Among the many significantly increased metabolites, we were highly interested in LysoPS, 7-α,25-Dihydroxycholesterol, and 15-d-PGJ2, which have a defined role in immunity and inflammation. Lysophosphatidylserine (LysoPS) is a well-known monoacyl derivative of phosphatidylserine (diacylPS), and possesses unique signaling characteristics important in acute inflammation and the orchestration of its resolution. Broadly, LysoPS has long been known as a signaling phospholipid in mast cell biology, enhancing histamine release and eicosanoid production. Further, there has been a resurgence of interest in LysoPS as it plays a crucial role in the resolution of inflammation by promoting efferocytosis [50,51,52]. Oxidant-derived LysoPS from neutrophils is known to be an endogenous anti-inflammatory mediator, mediating the “early” and rapid clearance of localized neutrophils [53]. LysoPS was found to block the activation of T cells via its interaction with its cognate receptor GPR174 to suppress IL-2 production by activated T cells and limit upregulation of the activation markers CD25 and CD69 [54]. We believe that EVs use LysoPS for constraining the local inflammation by neutralizing the activation of various immune cells, e.g., T cells, neutrophils, and mast cells, along with suppression of inflammatory cytokines [55] in targeted tissues during COVID-19 infection. However, this claim requires proper experimental validation.

7-α,25-Dihydroxycholesterol (7α, 25-OHC), an oxysterol, binds to EBI2 on immune cells. Recently published studies elucidated that 7α, 25-OHC bound to EBI2 regulates migration, activation, and functions of B cells, T cells, dendritic cells (DCs), monocytes/macrophages, and astrocytes [56]. In the majority of cell types, 7α, 25-OHC is derived from 25-hydroxycholesterol (25-HC) and possesses broad inhibitory activities against enveloped viruses of different families [57,58,59]. Though 25-HC has clearly shown strong antiviral activity against COVID-19 [60], direct evidence for the antiviral activity of its derivative 7α, 25-OHC is still lacking and requires further investigation. Nonetheless, our data suggest that EVs carry 7α, 25-OHC to position the B cells’ response in the target tissue and to block viral replication during COVID-19 infection.

Finally, we discuss 15-d-PGJ2, a potent and natural ligand of peroxisome proliferator-activated receptor-gamma (PPAR-γ), a member of the PPAR transcription factor family. Both natural (edible sources) and synthetic PPAR-γ agonists (e.g., rosiglitazone and pioglitazone) are considered to be one of the most promising candidates to improve the clinical outcome of viral diseases. These PPAR-γ agonists not only regress the inflammatory response to viral pneumonia but are also shown to promote the survival of influenza-infected mice [61,62,63,64]. For COVID-19-associated cytokine storm, PPAR-γ agonists are also in consideration [65].

## 4. Materials and Methods

### 4.1. Sampling, Ethical Aspects, and Patients

Samples were obtained from multi-ethnicity patients or donors residing in Saudi Arabia. All samples from COVID-19 patients were collected from individuals admitted to the hospital, based on meeting COVID-19 case definition as per the Saudi Ministry of Health (MOH) guidelines and confirmed by the RT-qPCR assay targeting the envelope (E) and RNA-dependent RNA polymerase (RdRp) genes. All samples were anonymized and used based on ethical approval obtained from the Unit of Biomedical Ethics in King Abdulaziz University Hospital (Reference No. 280-20) and the Institutional Review Board at the Ministry of Health, Saudi Arabia, with informed consent obtained from all participants. All methods and experiments were performed in accordance with the relevant guidelines and regulations. All healthy volunteers were confirmed with a PCR test and an antibody test to ensure that there was no history of infection. All the COVID-19 patients’ (n = 6, 3 in each group) and healthy controls’ (n = 3) samples were taken to KAU hospital between 1 July and 30 October 2020. In all cases, the included patients received the basic routine medical support established by the KAU hospital, that included the use of antibiotics, antivirals, vasopressor, anticoagulants, oxygen support, mechanic ventilation, and use of corticoids if necessary. The inclusion criteria included COVID-19 patients with mild to severe symptoms, while the exclusion criteria included COVID-19 patients without cancer or other viral infections such as HBV, HCV, HIV, and RSV.

The patients were divided into 2 groups, based on the course of the disease, as follows: the mild–severe group, which included patients with mild to severe symptoms that were discharged without admission to an ICU, and the severe-ICU group, which included patients with severe symptoms that were discharged after admission to an ICU. Healthy controls were randomly selected with no infection or other major diseases.

### 4.2. Exosomes Concentration and Characterization

At the time of hospital admission, blood samples were collected in vacutainer plain tubes for laboratory analyses. Serum was centrifuged at 3000× *g* for 15 min to remove cells and cell debris. The supernatant was then filtered through a 0.42 μm filter and incubated with ExoQuick (SBI CAT No. EXOQ20A-1, Embarcadero Way, Palo Alto, Canada) according to the manufacturing guide. Pellets were then resuspended in PBS and stored at –80 °C in a freezer. Nanoparticle tracking analysis (NTA) was used to count and size nanoparticles in suspension. Briefly, 10 µL of EXO suspension was loaded into the sample chamber of a Zetasizer Nano ZS90 (Malvern Panalytical, Malvern, U.K). Data analysis was performed using NTA software, version 3.3 (Malvern Panalytical, England). Transmission electron microscopy (TEM) was used to examine the morphology of the exosomes. Exosomes were fixed with 2.5% glutaraldehyde, then a drop of exosomes (10 μL) was added to a grid. All excess fluid was removed using filter paper, and the samples were negatively stained using an aqueous 1% phosphotungstic acid with pH 7.2. The samples were then air-dried and viewed using an electron microscope Titan CT_iac (FEI, Titan D3187).

### 4.3. Untargeted Metabolomics of EVs

#### 4.3.1. Metabolite Extraction

Extraction of metabolites from EVs was performed by adding some modifications to the pre-established protocols of our group [66,67,68], and untargeted metabolomics was performed using HPLC-tandem mass spectrometry (LC-MS/MS). All samples from each participant were analyzed in triplicate. Briefly, 400 µL of extracting solvent was mixed in ice-cold methanol:acetonitrile:chloroform:water at a ratio of 2:2:2:1 (*v*/*v*), added to the exosomes, followed by vortexing and incubation for an hour at –20 °C. Further, samples were then spun at 4 °C for 5 min at 8000 rpm, and supernatants were collected. Supernatant samples were analyzed using LC-MS/MS.

#### 4.3.2. HPLC Workflow

HPLC-based separation was performed with some modifications to the pre-established protocol: 10 µL of metabolite extract was injected into a HPLC column (Hypersail gold column C18 Part No: 25005-104630; 100 × 4.6 mm, 5 μm) with a flow rate of 0.25 mL/min. The mobile phase consisted of 0.1% formic acid and 99.9% ACN formic acid (0.1%, *v/v*), using a gradient program, where the component of solution was changed from 5% to 30% for 30 min, 30% to 50% for 10 min, 50% for 10 min, and finally, 50% to 95% for 20 min, at a constant flow rate, with a total running time of 70 min. The column temperature was maintained at 30 °C [69].

#### 4.3.3. Mass Spectrometry Parameters

A LTQ XL™ linear ion trap instrument (Thermo Fisher Scientific, USA) was used for the untargeted metabolomic analysis. Full scan scope was chosen from 80 to 1000 *m/z*. The spray voltage was set at −3.0 kV. The capillary voltage was fixed at 4.0 V and the temperature was set at 270 °C, collision energy 40 arbitrary units. Nitrogen was used as a sheath gas and the flow rate was set at 40 arbitrary units. Further, helium was used as the buffer gas for the run [70,71].

#### 4.3.4. Data Processing

The raw file data obtained from LTQ-XL were converted into mzXML using raw converter software. The mzXML files were processed using XCMS (https://xcmsonline.scripps.edu (accessed on 24 September 2021)) for feature detection, retention time correction, and alignment. The parameters in XCMS were set as follows: Cent-Wave settings for feature detection (Δ *m/z* = 30 ppm, minimum peak width = 10 s, and maximum peak width = 120 s) and *m/zwid* = 0.25, min frac = 0.5, and bw = 10 for chromatogram alignment. After careful evaluation of the retention time, alignment was shown to not be required. Isotopic peaks and adducts were detected using a camera. The precursor was matched with the METLIN database at 20 ppm accuracy.

#### 4.3.5. Metabolomics Pathway Analysis

The web-based R-package MetaboAnalyst 3.0 (https://www.metaboanalyst.ca/faces/home.xhtml (accessed on 24 September 2021)) was used to analyze metabolomics pathways [72].

### 4.4. Statistical Analysis

Statistical and chemometric analyses were performed as previously described [68]. Kruskal–Wallis non-parametric analysis was performed. Datasets were presented as box plot, median, and interquartile range. Statistical significance was defined as *p* ≤ 0.05.

## 5. Conclusions

Overall, our extensive untargeted metabolomic approach identified that exosomes not only carry active drug molecules but also anti-inflammatory metabolites, clearly suggesting that exosomes play a crucial role in negotiating with heightened inflammation during COVID-19 infection. However, one major limitation of our work is that we did not perform any functional validation regarding the anti-inflammatory activity of the three major metabolites during COVID-19 infection. We strongly believe that more future work will highlight the importance of exosomes as potential biological entities for the resolution of inflammation during COVID-19.

## Figures and Tables

**Figure 1 ijms-22-10467-f001:**
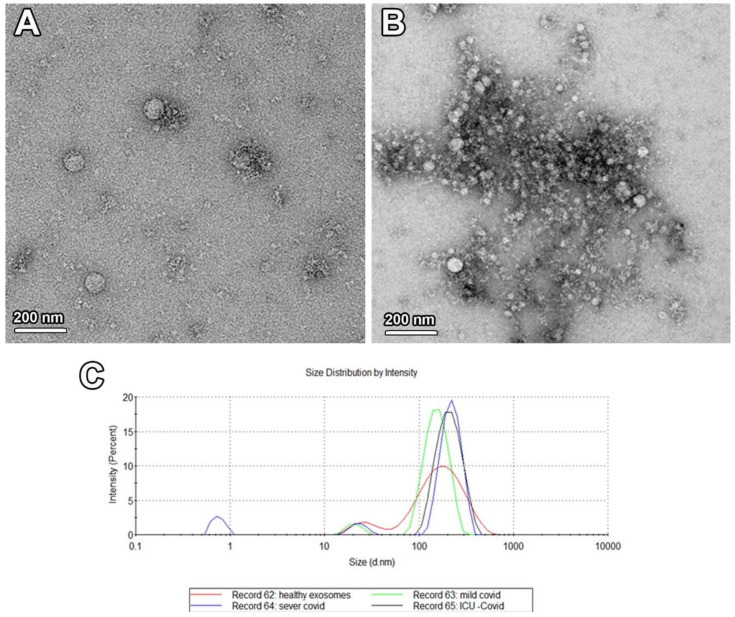
Characterization of isolated exosomes from control and patient samples. TEM examination shows MVs in serum samples isolated from the healthy controls (**A**) and COVID-19 patients (**B**). (**C**) NanoSight graph shows average size of the isolated exosomes.

**Figure 2 ijms-22-10467-f002:**
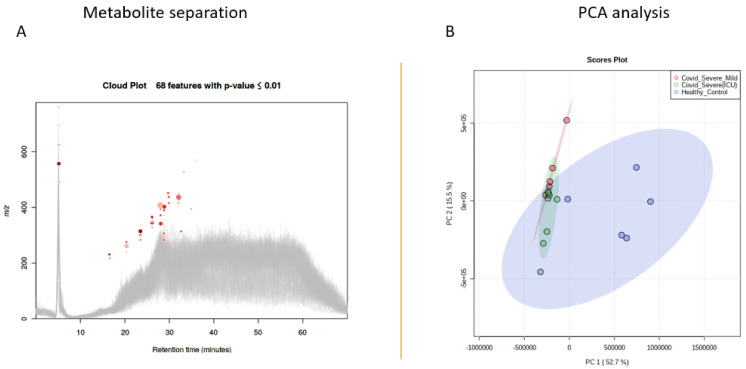
Metabolite separation and statistical analysis. (**A**) Total ion chromatogram of EVs isolated from both healthy and COVID-19 patient samples. (**B**) Scores plot between the healthy and COVID-19 patient samples. The explained variances are shown in brackets.

**Figure 3 ijms-22-10467-f003:**
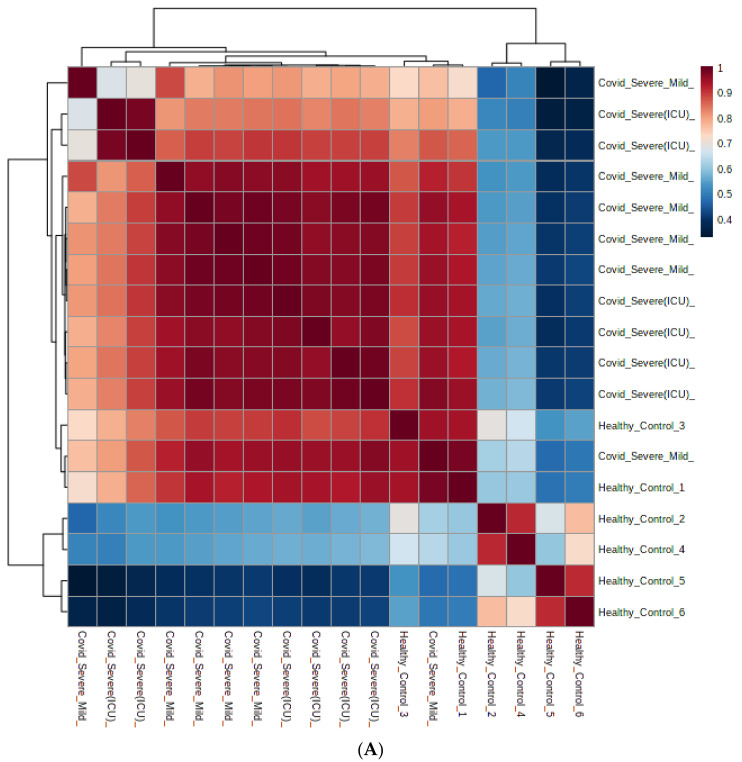
Correlation and expression of identified metabolites from both control and patient samples. (**A**) Correlation analysis of differentially modulated metabolites identified in both healthy and COVID-19 patient samples. (**B**) Clustering result is shown as a heatmap (distance measure using Euclidean distance and clustering algorithm using Ward heat map of individual samples).

**Figure 4 ijms-22-10467-f004:**
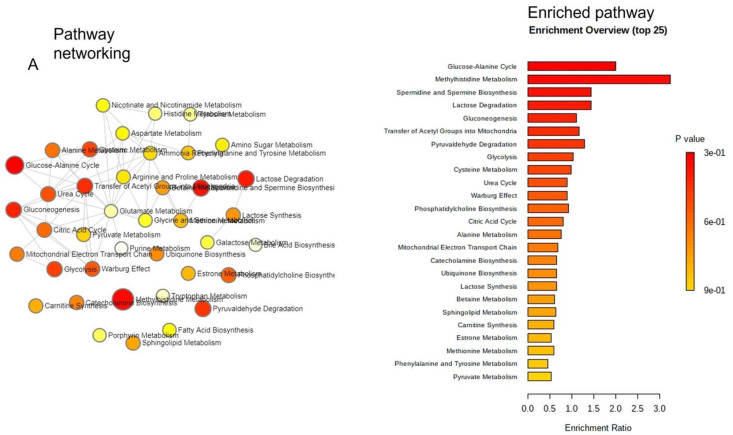
(**A**,**B**) Metabolite enrichment analysis: pathway networking, enriched pathways, and classification of accumulated lipid classes found in healthy control and COVID-19 patient EVs.

**Figure 5 ijms-22-10467-f005:**
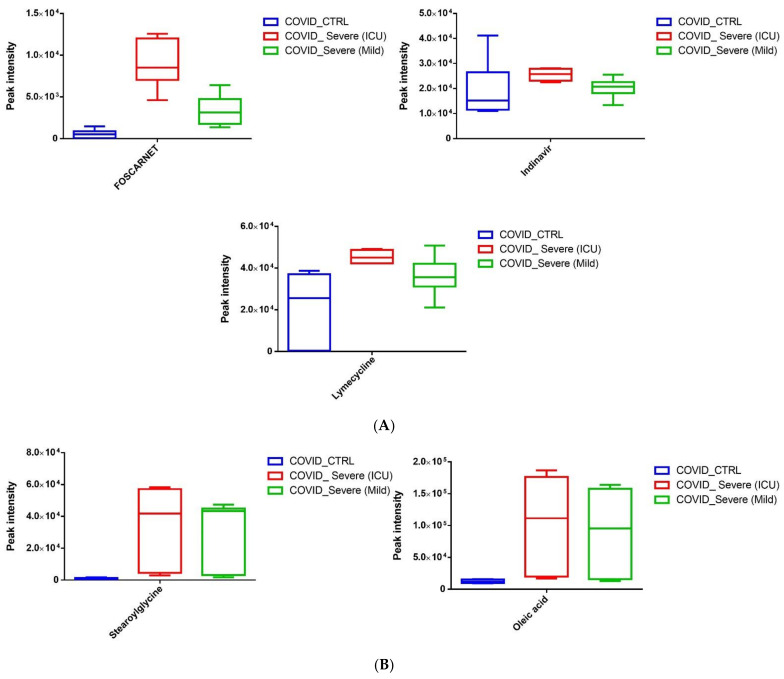
(**A**–**C**) Quantitative levels of statistically significant metabolites (*p* < 0.01) found to be upregulated in COVID-19 EVs when compared to healthy samples. Data are presented as box plot, median, and interquartile range (n = 3/group).

**Figure 6 ijms-22-10467-f006:**
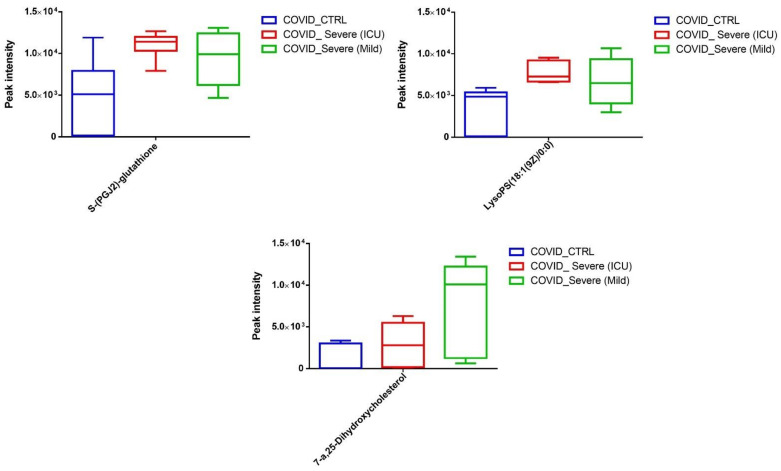
Quantitative levels of LysoPS, 7-α,25-Dihydroxycholesterol, and 15-d-PGJ2 in COVID-19 EVs when compared to healthy control samples.

**Figure 7 ijms-22-10467-f007:**
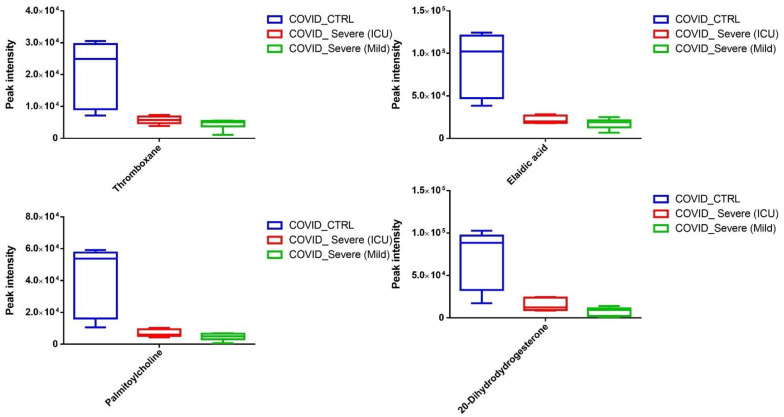
Quantitative level of statistically significant metabolites (*p* < 0.01) found to be downregulated in COVID-19 EVs when compared to healthy control samples. Data are presented as box plot, median, and interquartile range (n = 3/group).

**Figure 8 ijms-22-10467-f008:**
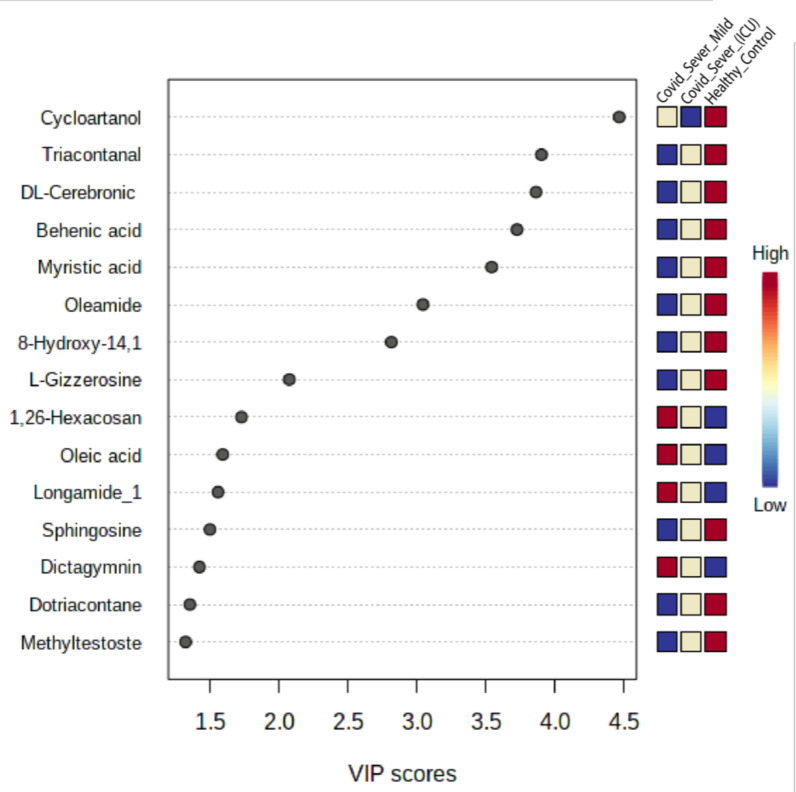
PLS-DA analysis for identification of crucial metabolites associated with disease. The colored boxes on the right indicate the relative concentration of the corresponding metabolites from EVs that are taken from both healthy and COVID-19 patient samples.

**Table 1 ijms-22-10467-t001:** Demographic data for COVID-19 patients (n = 3/group) and healthy control subjects (n = 3).

Group	Age	Gender	Main Medication(s)
Control	30.33 ± 2.40	3M	
Mild–Severe	54.33 ± 3.93	1M, 2F	Favipravir, Ceftriaxone, Azithromycin, Foscarnet, Indinavir, Lymecycline Dexamethasone, and Clexane
Severe-ICU	59.33 ± 7.69	2M, 1F	Favipravir, Ceftriaxone, Azithromycin, Foscarnet, Indinavir, and Lymecycline

M: Male and F: female.

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
