# Peer review of "Untargeted Metabolic Profiling of Extracellular Vesicles of SARS-CoV-2-Infected Patients Shows Presence of Potent Anti-Inflammatory Metabolites"

_ijms, 2021, doi:10.3390/ijms221910467_

Round 1

Reviewer 1 Report

Exploratory metabolomics remains important in elucidating pathogenic mechanisms of novel pathogens. I congratulate the authors for this detailed and well-written article. 

To allow readers better appreciation of the scientific merits, I would suggest the followings -

  1. Ln 33-35: "The results showed that antivirals and antibiotics like Foscarnet, Indinavir, and lymecycline were associated with EVs from patients treated with these drugs but were absent in EVs from untreated individuals"
    • care should be taken to avoid conclusions that are too obvious / tautological as they weaken the paper
    • rather than the "comparative" style, authors should just describe these expected, obvious findings
  2. Similarly, in the comparisons drawn, pp.8-9, 
  3. Sample labels: please rename Black and Blue to the severity grading used elsewhere in the article for consistency
  4. Figure 4 panel legends should be more tidy e.g. Pathway networking, Enriched pathway, Classification of lipid accumulation, Enriched lipids should be better aligned and consistently put on one (or two) lines
  5. Meaning of line 326-327 is ambiguous. If the authors are stating the supplier afterwards, it should be deleted. If they cannot state the supplier, they must have a justifiable reason. Simply saying "received from a commercial supplier" without naming the supplier is problematic. 
  6. Subheadings under section 4.3 should not be indented

Authors could also consider referring to the following articles as appropriate

Lipid metabolites as potential diagnostic and prognostic biomarkers for acute community acquired pneumonia.

To KK, Lee KC, Wong SS, Sze KH, Ke YH, Lui YM, Tang BS, Li IW, Lau SK, Hung IF, Law CY, Lam CW, Yuen KY.Diagn Microbiol Infect Dis. 2016 Jun;85(2):249-54. doi: 10.1016/j.diagmicrobio.2016.03.012. Epub 2016 Mar 14.PMID: 27105773    Lipid mediators of inflammation as novel plasma biomarkers to identify patients with bacteremia. To KK, Lee KC, Wong SS, Lo KC, Lui YM, Jahan AS, Wu AL, Ke YH, Law CY, Sze KH, Lau SK, Woo PC, Lam CW, Yuen KY.J Infect. 2015 May;70(5):433-44. doi: 10.1016/j.jinf.2015.02.011. Epub 2015 Feb 26.PMID: 25727996    

Author Response

Dear reviewer 1, 

Thank you very much for your time to review our manuscript trying to enrich it with your valuable remarks.

We fully understand your concern on certain points and we hope that we can give you a fair and satisfying explanation.

Exploratory metabolomics remains important in elucidating pathogenic mechanisms of novel pathogens. I congratulate the authors for this detailed and well-written article. 

To allow readers better appreciation of the scientific merits, I would suggest the followings -

  1. Ln 33-35: "The results showed that antivirals and antibiotics like Foscarnet, Indinavir, and lymecycline were associated with EVs from patients treated with these drugs but were absent in EVs from untreated individuals"
    • care should be taken to avoid conclusions that are too obvious / tautological as they weaken the paper
    • rather than the "comparative" style, authors should just describe these expected, obvious findings

Response 1: As per the suggestion of the learned reviewer, we have changed the language and now described the finding as “expected”.

  1. Similarly, in the comparisons drawn, pp.8-9, 

Response 2: As per the suggestion by the learned reviewer we have no corrected the language and described them as general findings.

  1. Sample labels: please rename Black and Blue to the severity grading used elsewhere in the article for consistency

Response 3: Done and the new names were applied throughout the whole manuscript.

  1. Figure 4 panel legends should be more tidy e.g. Pathway networking, Enriched pathway, Classification of lipid accumulation, Enriched lipids should be better aligned and consistently put on one (or two) lines

Response 4: As per the suggestion, we have modified the legend in Figure 4.

  1. Meaning of line 326-327 is ambiguous. If the authors are stating the supplier afterwards, it should be deleted. If they cannot state the supplier, they must have a justifiable reason. Simply saying "received from a commercial supplier" without naming the supplier is problematic. 

Response 5: Thanks for your comment. This sentence was deleted.

  1. Subheadings under section 4.3 should not be indented

      Response 6: Done.

  1. Authors could also consider referring to the following articles as appropriate

Lipid metabolites as potential diagnostic and prognostic biomarkers for acute community acquired pneumonia.

To KK, Lee KC, Wong SS, Sze KH, Ke YH, Lui YM, Tang BS, Li IW, Lau SK, Hung IF, Law CY, Lam CW, Yuen KY.Diagn Microbiol Infect Dis. 2016 Jun;85(2):249-54. doi: 10.1016/j.diagmicrobio.2016.03.012. Epub 2016 Mar 14.PMID: 27105773   

Lipid mediators of inflammation as novel plasma biomarkers to identify patients with bacteremia. To KK, Lee KC, Wong SS, Lo KC, Lui YM, Jahan AS, Wu AL, Ke YH, Law CY, Sze KH, Lau SK, Woo PC, Lam CW, Yuen KY.J Infect. 2015 May;70(5):433-44. doi: 10.1016/j.jinf.2015.02.011. Epub 2015 Feb 26.PMID: 25727996  

Response 7: As per the suggestion of the learned reviewer we have chosen the upper one to be added in the manuscript at relevant place.

Reviewer 2 Report

The bibliography in the article is not inserted as indicated by the journal.

I would explain a little bit more aspects about the Extracellular vesicles in the introduction.

At line 82 the authors state that "Several viruses utilize EVs to infect 82
host cells" could they please specify which of the viruses.

At line 100 the authors state that " Recent studies" and they cite just one study.

What were the inclusion and exclusion criteria of the patients in the study?

I would suggest the authors to cite also:
10.3390/microorganisms9030525

DOI

10.2147/RMHP.S284557

Author Response

Dear reviewer 2,

Thank you very much for your time to review our manuscript trying to enrich it with your valuable remarks. We fully understand your concern on certain points and we hope that we can give you a fair and satisfying explanation.

1) The bibliography in the article is not inserted as indicated by the journal.

Response 1: Thanks for your comment. All bibliography formats were corrected.

2) I would explain a little bit more aspects about the Extracellular vesicles in the introduction.

Response 2: We added some more details please see red text in Introduction.

3) At line 82 the authors state that "Several viruses utilize EVs to infect 82
host cells" could they please specify which of the viruses.

Response 3: Viruses were specified and we added some more details.

4) At line 100 the authors state that " Recent studies" and they cite just one study.

Response 4: Thanks for your comment. This was replaced by “A recent study has”

5) What were the inclusion and exclusion criteria of the patients in the study?

Response 5: Thanks for your comment. They were added.

6) I would suggest the authors to cite also:
10.3390/microorganisms9030525

DOI

10.2147/RMHP.S284557

Response 6: Thanks for your suggestion the first article (which is the most relevant to our topic) was cited in the manuscript.

Round 2

Reviewer 1 Report

Please fix the small grammatical errors and make consistent the capitalisation.